# Population Study of Ovarian Cancer Risk Prediction for Targeted Screening and Prevention

**DOI:** 10.3390/cancers12051241

**Published:** 2020-05-15

**Authors:** Faiza Gaba, Oleg Blyuss, Xinting Liu, Shivam Goyal, Nishant Lahoti, Dhivya Chandrasekaran, Margarida Kurzer, Jatinderpal Kalsi, Saskia Sanderson, Anne Lanceley, Munaza Ahmed, Lucy Side, Aleksandra Gentry-Maharaj, Yvonne Wallis, Andrew Wallace, Jo Waller, Craig Luccarini, Xin Yang, Joe Dennis, Alison Dunning, Andrew Lee, Antonis C. Antoniou, Rosa Legood, Usha Menon, Ian Jacobs, Ranjit Manchanda

**Affiliations:** 1Wolfson Institute of Preventative Medicine, Barts CRUK Cancer Centre, Queen Mary University of London, Charterhouse Square, London EC1M 6BQ, UK; f.gaba@qmul.ac.uk (F.G.); xintingliu@yahoo.co.uk (X.L.); s.goyal@smd15.qmul.ac.uk (S.G.); n.lahoti@smd15.qmul.ac.uk (N.L.); d.chandrasekaran@qmul.ac.uk (D.C.); 2Department of Gynaecological Oncology, St Bartholomew’s Hospital, Barts Health NHS Trust, London EC1A 7BE, UK; m.kurzer@nhs.net; 3School of Physics, Astronomy and Mathematics, University of Hertfordshire, College Lane, Hatfield AL10 9AB, UK; o.blyuss@qmul.ac.uk; 4Department of Paediatrics and Paediatric Infectious Diseases, Sechenov First Moscow State Medical University, Moscow 119146, Russia; 5Department of Applied Mathematics, Lobachevsky State University of Nizhny Novgorod, Nizhny Novgorod 603098, Russia; 6Department of Women’s Cancer, Elizabeth Garrett Anderson Institute for Women’s Health, University College London, London WC1E 6AU, UK; j.k.kalsi@ucl.ac.uk (J.K.); a.lanceley@ucl.ac.uk (A.L.); 7Department of Behavioural Science and Health, University College London, 1-19 Torrington Place, London WC1E 6BT, UK; saskia.sanderson@ucl.ac.uk; 8Department Clinical Genetics, North East Thames Regional Genetics Unit, Great Ormond Street Hospital, London WC1N 3JH, UK; munaza.ahmed@gosh.nhs.uk; 9Department of Clinical Genetics, University Hospital Southampton NHS Foundation Trust, Southampton SO16 6YD, UK; Lucy.Side@uhs.nhs.uk; 10Medical Research Council Clinical Trials Unit at UCL, Institute of Clinical Trials and Methodology, University College London, 90 High Holborn, London WC1V 6LJ, UK; a.gentry-maharaj@ucl.ac.uk (A.G.-M.); u.menon@ucl.ac.uk (U.M.); 11West Midlands Regional Genetics Laboratory, Birmingham Women’s NHS Foundation Trust, Birmingham B15 2TG, UK; y.wallis@nhs.net; 12Manchester Centre for Genomic Medicine, 6th Floor Saint Marys Hospital, Oxford Rd, Manchester M13 9WL, UK; Andrew.Wallace@mft.nhs.uk; 13Cancer Prevention Group, King’s College London, Great Maze Pond, London SE1 9RT, UK; jo.waller@kcl.ac.uk; 14Centre for Cancer Genetic Epidemiology, Department of Public Health and Primary Care, University of Cambridge, Strangeways Laboratory, Worts Causeway, Cambridge CB1 8RN, UK; craig@srl.cam.ac.uk (C.L.); xy249@medschl.cam.ac.uk (X.Y.); jgd29@cam.ac.uk (J.D.); amd24@medschl.cam.ac.uk (A.D.); ajl65@medschl.cam.ac.uk (A.L.); aca20@medschl.cam.ac.uk (A.C.A.); 15Department of Health Services Research and Policy, London School of Hygiene and Tropical Medicine, London WC1E 7HT, UK; Rosa.Legood@lshtm.ac.uk; 16Department of Women’s Health, University of New South Wales, Australia, Level 1, Chancellery Building, Sydney 2052, Australia; i.jacobs@unsw.edu.au

**Keywords:** population genetic testing, ovarian cancer risk, risk stratification, *BRCA1*, BRCA2, RAD51C, RAD51D, BRIP1, SNP, risk modelling

## Abstract

Unselected population-based personalised ovarian cancer (OC) risk assessment combining genetic/epidemiology/hormonal data has not previously been undertaken. We aimed to perform a feasibility study of OC risk stratification of general population women using a personalised OC risk tool followed by risk management. Volunteers were recruited through London primary care networks. Inclusion criteria: women ≥18 years. Exclusion criteria: prior ovarian/tubal/peritoneal cancer, previous genetic testing for OC genes. Participants accessed an online/web-based decision aid along with optional telephone helpline use. Consenting individuals completed risk assessment and underwent genetic testing (*BRCA1/BRCA2/RAD51C/RAD51D/BRIP1*, OC susceptibility single-nucleotide polymorphisms). A validated OC risk prediction algorithm provided a personalised OC risk estimate using genetic/lifestyle/hormonal OC risk factors. Population genetic testing (PGT)/OC risk stratification uptake/acceptability, satisfaction, decision aid/telephone helpline use, psychological health and quality of life were assessed using validated/customised questionnaires over six months. Linear-mixed models/contrast tests analysed impact on study outcomes. Main outcomes: feasibility/acceptability, uptake, decision aid/telephone helpline use, satisfaction/regret, and impact on psychological health/quality of life. In total, 123 volunteers (mean age = 48.5 (SD = 15.4) years) used the decision aid, 105 (85%) consented. None fulfilled NHS genetic testing clinical criteria. OC risk stratification revealed 1/103 at ≥10% (high), 0/103 at ≥5%–<10% (intermediate), and 100/103 at <5% (low) lifetime OC risk. Decision aid satisfaction was 92.2%. The telephone helpline use rate was 13% and the questionnaire response rate at six months was 75%. Contrast tests indicated that overall depression (*p* = 0.30), anxiety (*p* = 0.10), quality-of-life (*p* = 0.99), and distress (*p* = 0.25) levels did not jointly change, while OC worry (*p* = 0.021) and general cancer risk perception (*p* = 0.015) decreased over six months. In total, 85.5–98.7% were satisfied with their decision. Findings suggest population-based personalised OC risk stratification is feasible and acceptable, has high satisfaction, reduces cancer worry/risk perception, and does not negatively impact psychological health/quality of life.

## 1. Introduction

*BRCA1/BRCA2* pathogenic variants have a 17–44% ovarian cancer (OC) risk until age 80 years [1]. Testing for OC susceptibility genes (CSGs)—*RAD51C* (lifetime OC risk = 11%) [2], *RAD51D* (lifetime OC risk = 13%) [2] and *BRIP1* (lifetime OC risk = 5.8%) [3]—is now part of clinical practice. Genome-wide association studies (GWAS) have discovered ~30 validated single-nucleotide polymorphisms (SNPs) which modify OC risk [4,5]. Newer risk prediction models incorporating validated SNPs as a polygenic risk score with epidemiologic/family history(FH)/hormonal data and moderate–high-penetrance CSGs can be used to predict lifetime OC risk, improving the precision of risk estimation and allowing population division into risk strata, enabling targeted downstream risk-stratified prevention/screening for those at increased risk [4,6].

The current practice of identifying high-risk women uses clinical criteria/FH-based testing for CSGs, misses >50% CSG carriers who do not fulfil genetic testing criteria and requires people to get cancer before identifying unaffected family members who can benefit from prevention [7,8,9,10]. Given the effective cancer risk management/prevention options available, the adequacy of current practice, representing massive missed opportunities for risk-stratified prevention, is questionable. Unselected population genetic testing (PGT) overcomes these limitations and identifies many more individuals at increased OC risk. PGT can be cost effective and prevent thousands of more OC/BC cases than clinical criteria/FH-based genetic testing [11].

Most PGT evidence comes from UK/Israeli/Canadian studies in Ashkenazi Jewish (AJ) populations [9,10,12]. These show that AJ population-based *BRCA* testing is acceptable, feasible, can be community based, doubles the *BRCA* pathogenic variant individuals identified, does not harm psychological health/quality of life (QoL), reduces long-term anxiety, has high satisfaction rates (90–95%) [9,10,13], and is extremely cost effective (potentially cost saving) for the UK/US health systems [14]. However, prospective/unbiased PGT data and model-based OC risk stratification for a general (non-Jewish) low-risk population are lacking. 

We describe results from a feasibility study in order to stratify a general population using predicted lifetime OC risk and offer risk management options of screening and prevention, within the Predicting Risk of Ovarian Malignancy Improved Screening and Early detection programme (PROMISE-FS, ISRCTN54246466). This article reports on (1) the acceptability, feasibility, and uptake of PGT/OC risk stratification; (2) perceived risks/limitations; (3) decision aid (DA)/telephone helpline use; (4) satisfaction; (5) cancer worry/risk perception; (6) impact on psychological health/QoL.

## 2. Results

Between June 2017 and August 2017, 218 women registered and 123 viewed the online DA. In total, 105/123 (85%) DA users consented to genetic testing/risk assessment, and two withdrew. In total, 103 were eligible for analysis (Figure 1). In total, 2/103 were excluded from RPA assessment (Figure 1). Women who chose not to participate declined providing information on factors affecting decision making. The follow-up questionnaire response rate was 94%, 84%, and 75% at seven days, three months and six months post results, respectively.

Table 1 summarises cohort baseline characteristics. The mean age of participants was 48.5 (SD = 15.4; range = 18–85) years; 44.6% (*n* = 45) had university level education; 55.7% (*n* = 54) had a household income >£40,000; 74.5% (*n* = 76) were Caucasian; 7% (*n* = 7) were smokers; 64% (*n* = 63) ate >5 portions of fruit/vegetables daily; 78% (*n* = 80) were physically active over the last month. None had a clinically significant FH of cancer (fulfilling NHS genetic testing criteria). RPA revealed 1/103 at ≥10%, 0/103 at ≥5%–<10% and 100/103 at <5% lifetime OC risk. As expected using the algorithm, the epidemiological risk factors alone provide a greater level of OC risk stratification among the participants compared to the polygenic risk score (PRS) alone (Appendix A). However, risk stratification is further improved when the full model incorporating both epidemiological risk factors and PRS is considered. One high-risk participant, aged 35 years, had a lifetime OC risk of 42%. She had a pathogenic duplication of exon-13 in BRCA1. History included one second-degree relative with OC—parity = 1, 10 years oral contraceptive pill (OCP) use, endometriosis, BMI = 30.4, and no tubal ligation/hormone-replacement therapy (HRT) use. Following results, the participant opted for Risk of Ovarian Cancer Algorithm (ROCA)-based screening (24) within a research study (ALDO, https://www.uclh.nhs.uk/OurServices/ServiceA-Z/Cancer/NCV/Pages/TheALDOproject.aspx) and for risk-reducing early salpingectomy within a clinical trial (PROTECTOR, ISRCTN25173360, http://www.protector.org.uk/). She underwent MRI screening for BC risk. Four Class-3 variants of uncertain significance (VUS) were detected (BRCA1:c.3328_3330delAAG, c.2998_3003del; BRCA2:c.1438T>G; RAD51D:c.482T>C).

Key perceived benefits/risks of PGT/OC risk assessment are shown in Appendix A. Need for reassurance, reduction in uncertainty, enhancing cancer prevention, benefiting research, knowledge about enhanced screening/prevention and children’s risks were rated somewhat/very important by ~70–98% women. Important risks/limitations of PGT/OC risk assessment rated somewhat/very important included concern about effect on family (56.4%) and being unable to handle it emotionally (38.6%). A minority felt stigmatization (9%) or targeting of an ethnic group (11%) was a somewhat/very important risk. Insurance and confidentiality were highlighted as somewhat/very important by 28% and 24.7% respectively.

Participant responses to the ten DA items are shown in Appendix A. The mean number of times DA was viewed was non-significantly higher in consenters versus decliners (1.61 vs. 1.05; *p* = 0.06). The mean DA score was not significantly different between consenters and decliners (8.1 vs. 7.4; *p* = 0.14). Consenters were older than decliners (48.5 vs. 40, *p* = 0.016). The mean age of volunteers who registered but did not view the DA was 45.5 years and not significantly different from consenters (*p* = 0.16) or decliners (*p* = 0.24). There was no statistically significant difference in 9/10 DA item responses between consenters and decliners. (Appendix A). In total, 88.3% of consenters versus 75% of decliners (*p* = 0.036) would regret not participating if they developed OC in the future. In total, 23/123 viewed the DA on multiple occasions, and DA scores increased on repeat attempts (Appendix A). For 122/123 participants, there was concordance between participant decision making and DA outcome category. One participant (85 years, Caucasian, no OC-FH) consented to PGT/OC risk stratification despite DA advice to the contrary (DA score = −1). Table 2 summarises responses to the DA evaluation questionnaire. In total, 92.2% (94/102) were very satisfied/satisfied and 82.2% (83/101) would recommend the DA. The amount of information provided, length of time taken to view and level of detail available was deemed just right by 98% (100/102), 97.1% (99/102), and 97% (98/101), respectively. No part of the DA needed omitting.

In total, 13% (13/103) of consenters used the optional telephone helpline (Table 3), and 8/13 filled in an evaluation questionnaire. No decliner used the telephone helpline. The mean number of calls to the telephone helpline was 1.38 (SD = 1.12; range = 1–5). In total, 12.5% (1/8) used the telephone helpline to aid decision making and 75% (6/8) had study specific queries—of which, DA technical assistance queries (4/8) were the most common. All helpline users were very satisfied/satisfied with their experience and 75% (6/8) would recommend the helpline. In total, 37.5% (3/8) felt that the helpline aided decision making. There was no difference in baseline characteristics between helpline users and non-users. When comparing how much the DA improved understanding of OC/gene testing/advantages and disadvantages of discovering personalised OC risk or DA satisfaction, there was no statistically significant difference between helpline users/non-users. Helpline users had a significantly greater degree of worry (2/13 vs. 0/89; *p* = 0.02) and upset (1/13 vs. 0/89; *p* = 0.003) when viewing the DA in comparison to non-users. Helpline users had a higher DA mean score than non-users (9.123 vs. 8.019; *p* = 0.032)

Mean Hospital Anxiety and Depression Scale (HADS)/EuroQol-5D-5L (EQ-5D-5L)/Impact of Events Scale (IES)/Cancer Risk Perception questionnaire (CRP)/Cancer Worry Scale questionnaire (CWS)/Decision Regret Satisfaction questionnaire (DRS) questionnaire scores at baseline and at seven days/three months/six months follow up are shown in Table 4. 

Linear random-effects mixed-model outputs showing the association of covariates with different outcomes are shown in Table 5. There was a transient increase in HADS anxiety at seven days (*p* = 0.048), returning to baseline by three months (*p* = 0.318). Compared to baseline, there was a small increase in HADS depression scores at individual time points of 3 months (*p* = 0.027) and 6 months (*p* ≤ 0.001), while QoL scores were marginally lower at three (*p* = 0.025) and six months (*p* = 0.036). However, the absolute level of change from baseline in all these scores was extremely small (HADS depression = 2.92 to 3.55; HADS anxiety = 6.11 to 7.02; EQ-5D-5L = 0.86 to 0.84) and not clinically meaningful. Additionally, contrast tests evaluating whether overall mean values at seven days, three months and six months were jointly different from the baseline suggested that anxiety, depression and QoL at these time points were not jointly different from the baseline value for the cohort (Table 5). Distress scores decreased with time and were significantly lower at six months versus 7 days (*p* = 0.042). Compared to baseline, OC worry was significantly lower at 7 days (*p* ≤ 0.001), 3 months (*p* ≤ 0.001) and 6 months (*p* ≤ 0.001). Contrast tests evaluating the overall time effect showed a significant decrease in OC worry scores (*p* = 0.02) but not distress scores (*p* = 0.25) over time (Table 5). General cancer risk perception showed a decrease at 7 days (*p* = 0.012), returning to baseline by 6 months (*p* = 0.45).

In total, 85.5% strongly agreed and 13.2% agreed that their decision to undergo PGT/OC risk stratification was the right decision and that they were satisfied with it. In total, 95% would make the same choice again. Only 1.3% regretted their decision. Table 6 summarises responses to the DRS questionnaire.

A FH of BC (*p* = 0.034) but not OC (*p* = 0.20) was negatively associated with QOL. Having a FH of OC was not associated with an increase in OC worry or general cancer risk perception. However, women with a FH of BC perceived themselves to be at higher cancer risk (*p* = 0.002) but did not have increased OC worry.

Results from contrast tests assessing the joint effect of between-group and within-group differences in various outcomes over six months compared to baseline are shown in Table 7. There was no statistically significant between-group difference for groups ‘with’ and ‘without’ a FH of OC for HADS total/HADS depression/HADS anxiety/QoL/distress/OC worry/general cancer risk perception over time. There was no statistically significant within-group difference for groups ‘with’ and ‘without’ a FH of OC for HADS total/HADS anxiety/QoL/general cancer risk perception over six months. However, there was a statistically significant within-group difference for individuals ‘without’ a FH of OC but not ‘with’ a FH of OC for HADS depression (*p* = 0.003, *p* = 0.866, respectively), distress (*p* = 0.043, *p* = 0.524 respectively) and OC worry (*p* ≤ 0.001, *p* = 0.582, respectively) over six months. Viewing the contrast tests together in combination with the linear random-effects mixed-model outputs would suggest a small increase in HADS depression scores not of clinical significance and a decrease in distress and OC worry over six months for the ‘without’ a FH of OC group.

## 3. Discussion

This is the first unselected population-based, prospective cohort study recruiting participants without cancer history in self/family, evaluating the feasibility of personalised lifetime OC risk stratification followed by offering risk management options. Data suggest that OC risk stratification using genetic/non-genetic (epidemiological/hormonal) factors in general population women is feasible and acceptable.

The 85% uptake of PGT and OC risk stratification suggests high acceptability, similar to previously published data indicating putative 85% uptake of PGT (*n* = 734/829 in a survey study assessing attitudes of a general population of women to unselected PGT and risk-stratified OC screening [15,16]. Findings are also similar to data showing the high acceptability of unselected *BRCA* testing in AJ populations (up to 88% uptake) [17]. The 85%–98% overall satisfaction we found with PGT is similar to rates reported with population-based *BRCA* testing in AJ populations [9,12].

Data from unselected *BRCA* testing in the AJ population [9,10,14,18,19] show acceptability/feasibility/effectiveness/cost effectiveness/lack of detrimental impact on psychological health/QoL, and support the concept of population-based *BRCA* testing in Jewish populations. However, these inferences cannot be directly generalized to a non-Jewish general population. Our findings of overall time effect contrast tests showing levels of anxiety/depression/QoL/distress not being jointly different from baseline values but a significant reduction in OC-specific worry/general cancer risk perception following OC risk stratification are reassuring. Small changes in scores observed in some outcomes at individual time points were not clinically meaningful. While larger studies are warranted, these initial findings concur with short-/long-term outcome data following unselected *BRCA* testing in AJ populations [9,13] and are similar to findings amongst high-risk individuals undergoing clinical criteria-based genetic testing [20,21,22]. In total, 25.5% of our cohort was non-Caucasian (13.7% Asian). We found no difference in psychological health/QoL outcomes amongst non-Caucasians versus Caucasians. More research is required for understanding the role of various risk factors in non-Europeans.

Our online DA was successfully completed by women from a wide range of ages (18–85), education levels, and ethnicities, with high levels (92.2%) of satisfaction. Women who used the optional telephone helpline reported higher levels of worry/upset when viewing the DA. In total, 75% of women using the telephone helpline did so for technical DA assistance. All went on to successfully view the online DA. The telephone helpline appears to have been used as a source of emotional/technical support, emphasising the importance/need for a telephone helpline as an adjunct to online web applications to facilitate access/decision making for PGT/OC risk stratification. That one volunteer consented despite her DA score (−1) indicating she was “leaning against taking part”, highlights that whilst decision aids are adjuncts aiding decision making, individuals retain ultimate autonomy. While we showed the feasibility of using an online DA and helpline approach for PGT, this has not been compared in randomised trials to more standard/established methods (face-to-face/telephone-based/DVD-assisted counselling).

Our study strengths include population-based recruitment in a non-Jewish, ethnically diverse general population. We engaged and worked with primary care networks prior to study commencement. They helped increase awareness of this study, identify eligible women and facilitate recruitment. Engagement with primary care would be vital for the implementation of any national population-based model for PGT/OC risk stratification. Other advantages include a good questionnaire response rate, ranging from 99% (baseline) to 75% (six months follow up).

Limitations include the small sample size, lack of long-term follow up on QoL/psychological health/health behaviours. Additionally, this study was non-randomised and a control arm (without genetic testing) to compare any change in outcomes was lacking. However, the high-risk individual identified did opt for appropriate screening and preventive interventions to reduce OC/BC risk. Lack of intermediate-risk women identified probably reflects the small sample size.

In our cohort, 45% vs. 40% [23] of the UK general population had a university level education; 7% vs. 15% [23] were current smokers; 64% vs. 32% [23] ate the recommended ≥5 portions of fruit/vegetables daily; 78% vs. 64% [23] were physically active over the last month; median total household income was >£50,000 vs. £29,000 in the UK general population [23]. Higher income, education levels and healthy lifestyle behaviour in our study participants compared to the UK’s general population may indicate a London bias. The income/education levels/lifestyle choices are similar to those of the UK Jewish population [9,17]. Significant associations of some study outcome variables seen with demographic variables of income/age are consistent with observations from population-based data reported in other population cohorts.

Precision prevention is a prevention strategy incorporating individual variation in genetic, epi-genetic and non-genetic (e.g., environment, hormonal, lifestyle, behavioural) risk factors. This comprises primary prevention to prevent occurrence of disease and, secondary prevention for screening/early detection of pre-symptomatic disease. Next-generation sequencing technologies, falling costs and advances in computational bioinformatics makes personalised risk-stratified prevention feasible. Improvements in the precision of risk estimation, genetic understanding of disease and increasing awareness offers an opportunity to apply this knowledge and technology at a broad population scale to make an important shift in health care towards disease prevention. Over 50% of OCs occur in 9% of the population, which is at >5% OC risk [4]. This provides a huge opportunity for population stratification for precision prevention. Identification of unaffected women at increased risk offers opportunities for risk-stratified prevention to reduce cancer burden. Women at increased OC risk can opt for risk-reducing salpingo-oophorectomy (RRSO) to prevent tubal cancer/OC [24], now advocated at >4–5% lifetime OC risk [25,26,27].

Access to and uptake of testing for CSGs remains restricted. Only a small proportion of at-risk *BRCA* carriers have been identified [7,8]. Our approach offers opportunities to maximise pathogenic variant identification and population stratification for OC prevention. While recent data suggest that population-based genetic testing for OC/BC gene pathogenic variants could be cost effective in general population women [11], additional research including general population implementation studies are needed to address knowledge gaps before considering this. Additional looked for findings have recently been offered and returned following post hoc sequencing and/or analysis of some large genomic study cohorts. These studies would enable evaluation of CSG pathogenic variant carriage rates. However, this would not address in a prospective unbiased fashion key questions around the (i) logistics of population testing; (ii) information giving, a priori informed consent, and uptake of testing; (iii) uptake of preventive options. This ‘bolt-on’ paradigm of returning additional ‘secondary findings’ cannot be equated to prospective uptake of testing CSGs in an unselected unaffected population.

A prospective, Canadian cohort study offering *BRCA1/BRCA2* testing to unselected men/women (The Screen Project) is ongoing. The study is evaluating the feasibility of a direct-to-consumer approach, satisfaction, OC worry, prevalence of *BRCA1/BRCA2* pathogenic variants and the number of OCs/BCs prevented. Results from our feasibility study would inform the development of a larger UK-wide study that implements PGT/OC risk-stratified prevention. An important challenge is identifying optimum implementation pathways. It is likely that different context-specific models are needed for various health care systems internationally. Risk assessment pathways could be established through a community/primary care-based approach outside the traditional hospital-based genetics clinic model. A key issue that needs resolving is a system for monitoring/managing VUS. Commissioning/funding of a system where laboratory reports can be reviewed and re-issued in light of new evidence is needed. A framework/structure for data management and legal and regulatory protections will also need to be established.

## 4. Materials and Methods

### 4.1. Design

A multicentre, prospective cohort, feasibility study (ISRCTN:54246466). Inclusion criteria: women ≥18 years. Exclusion criteria: history of ovarian/tubal/primary peritoneal cancer or previous genetic testing for OC CSGs.

### 4.2. Recruitment

Recruitment was by self-referral. Study information/leaflets were made available through North-East London primary care practices. Interested volunteers received a detailed participant information sheet and access to an online DA prior to consent to genetic testing/participation. All had access to use an ‘optional’ telephone helpline for support/advice/queries. The helpline was manned by a doctor/research nurse experienced in cancer genetic risk assessment/management. Individuals deciding to undergo PGT/OC risk assessment consented. Decliners were asked to provide information on factors affecting decision making.

### 4.3. Decision Aid (DA)

A bespoke web-based DA was developed, enabling potential participants to make an informed decision on whether they wish to determine their OC risk and undergo PGT/OC risk assessment [16,28]. The DA (Appendix A) included information on OC, genetic testing and the PROMISE programme, followed by ten questions/items on potential advantages/disadvantages of learning about OC risk. Responses were rated according to two different 3-point Likert scales. Individual questions were scored according to responses ((a) 1 = in favour of taking part, −1 = against taking part, 0 = neither in-favour or against taking part; or (b) 1 = agree, −1 = disagree, 0 = unsure). Sum of all questions/items scores taken together ranged from −10 to 10. Women with total scores between −10 and −1 were considered “leaning against taking part”, 0–5 “undecided”, and 6–10 “leaning towards taking part”.

### 4.4. Genetic Analysis

Genetic testing involved next-generation sequencing of *BRCA1/BRCA2/RAD51C/RAD51D*/*BRIP1* genes and 30 GWAS-validated OC SNPs. Pathogenic variants detected were reconfirmed in an NHS laboratory.

### 4.5. Risk Model

Epidemiological/hormonal/reproductive data affecting OC risk collected at baseline (age/OC-FH/body mass index (BMI)/tubal ligation/hormone-replacement therapy (HRT)/oral contraceptive pill (OCP) use/endometriosis/parity) were combined with genetic information in a risk prediction algorithm (RPA) to provide a personalised predicted lifetime OC risk (till 80 years). Model validation (personal communication) [5] was undertaken in prospective datasets and cancers accrued in the UK OC screening trial cohorts [5,29,30]. Following RPA assessment, all participants were stratified into risk categories by lifetime OC risk (low risk: <5%; intermediate risk: ≥5%–<10%; high risk: ≥10%).

### 4.6. Test Result Management

High/intermediate-risk (and an equivalent number of randomly selected low-risk) individuals received their result at a face-to-face post test risk stratification counselling appointment. Identified pathogenic/likely pathogenic variant heterozygotes were referred to an NHS regional genetics clinic for confirmatory testing and to established NHS risk management services. Other low-risk individuals received results via post. Variants of uncertain significance (VUS) results were not returned.

### 4.7. Assessment of Demographics, Outcomes and Follow Up

Sociodemographic, family history, perceived risk/limitation (4-point Likert scale), telephone helpline and DA evaluation data were collected using customised questionnaires. Anxiety and depression were assessed with the Hospital Anxiety and Depression Scale (HADS) [31]. Distress was assessed using the Impact of Events Scale (IES) [32]. General cancer risk perception was measured by two items. Comparative risk: ‘Compared with other people of your age/sex, do you think your chances of getting cancer in your life are: much-lower, lower, about-the-same, higher, much-higher?’ An additional risk item: ‘On a scale from 0 to 100, where 0 is no chance at all and 100 is absolutely certain, what are the chances you will get cancer sometime during your lifetime?’. OC worry was assessed by the Cancer Worry Scale (CWS) [33]. Generic QoL was measured with the EQ-5D-5L questionnaire [34]. Satisfaction and regret were measured by the Decision Regret Satisfaction Scale (DRS) and one additional 5-point Likert scale item, ‘I am satisfied with the decision I have made’ [35]. Smoking, diet and physical activity were evaluated. Data were gathered at baseline following consent and post results delivery (seven days/three months/six months).

### 4.8. Statistical Analysis

Descriptive statistics were used for baseline characteristics/telephone helpline/DA/follow-up questionnaire data. The Wilcoxon rank sum test and Fisher’s exact test evaluated differences in means and proportions correspondingly.

As outcome data from the HADS/EQ-5D-5L/IES/CWS/CRP/DRS questionnaires were collected over multiple time points, linear random-effects mixed models were used to allow for individual baseline-level variability. Each scale was analysed, with the outcome as a continuous response variable. Models included a group effect and time effect. Models were adjusted for FH of OC/BC (positive/negative), age, income (in £10,000 increments), marital status (cohabiting/living with partner/married versus divorced/separated/single/widowed), ethnicity (Caucasian versus non-Caucasian) and religion (Muslim/Christian/Jewish/no religion/other (Hindu/Buddhist/Sikh)). Linear random-effects mixed models were used to model trends in DA scores for participants viewing the DA on multiple occasions.

Post modelling, three contrast tests were considered (each on three degrees of freedom). We assessed (a) overall time effects, i.e., whether the overall mean values at seven days, three months and six months from baseline were jointly different from the baseline level, (b) between-group differences over time (whether the mean group differences between those ‘with’ and ‘without’ a FH of OC at seven days, three months and six months from baseline were jointly different from the baseline level) and (c) within-group differences over time (whether mean values at seven days, three months and six months from baseline were jointly different from the baseline level within the groups ‘with’ and ‘without’ FH of OC). Statistical analysis used Stata-13.0 (Stata-Corp-LP, TX, https://www.stata.com/) and R version 3.5.1 (https://www.r-project.org/).

## 5. Conclusions

Our current health care systems remain primarily centred on improving disease diagnosis and treatment rather than prevention. Prevention of chronic disease, cancer being the second most common cause, is a major challenge for our health systems. PGT and personalised OC risk stratification can spur CSG detection and maximise precision prevention to reduce OC burden. We have shown that population-based personalised OC risk stratification is feasible and acceptable, has high satisfaction, reduces cancer worry/risk perception, and does not negatively impact on psychological health/quality of life. Further research and implementation studies evaluating the impact, clinical efficacy, long-term psychological, and socioethical consequences and cost effectiveness of this strategy are needed. This includes evaluation through large implementation studies of real-world health outcomes. Future implementation of such a strategy will require varying levels of workforce expansion/upskilling and reorganisation of health service infrastructure covering aspects of genetic testing and downstream care including screening and prevention pathways. PGT is an exciting and evolving field and personalised OC risk stratification offers a new paradigm for precision prevention in OC.

## Figures and Tables

**Figure 1 cancers-12-01241-f001:**
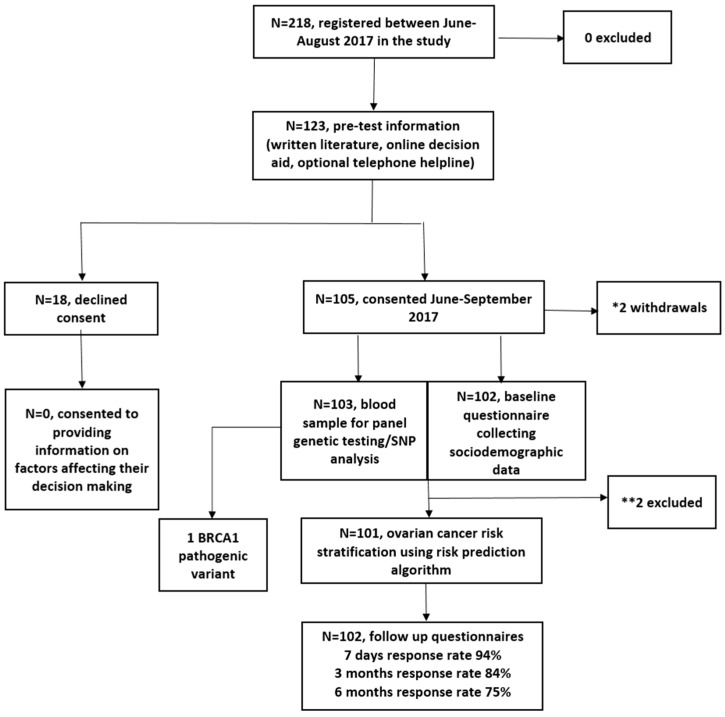
Study flow chart. * Reasons for withdrawal: miscarriage (*n* = 1) and inability to use public transport to attend an outpatient blood test appointment for genetic testing (*n* = 1). ** Reasons for exclusion: one participant was excluded because she entered this study at age 84 and the model predicts risks to age 80; the second participant did not provide the baseline demographic information required to run the algorithm.

**Table 1 cancers-12-01241-t001:** Baseline characteristics of cohort.

Characteristic	%	N
**Mean age, years (SD, range)**		48.54 (15.42, 18–85)
**Marital status**		
Single	20	20/100
Married	52	52/100
Cohabiting/living with partner	15	15/100
Divorced/separated	11	11/100
Widowed	2	2/100
**Children**		
Have children	65.69	67/102
Mean number of children (SD, range)		1.34 (1.23, 0–5)
**Education**		
No formal qualification	5.94	6/101
GCSE/O-level/CSE	17.82	18/101
NVQ1/NVQ2	0.99	1/101
A-level/NVQ3	26.73	27/101
NVQ4	3.96	4/101
Bachelors	26.73	27/101
Masters	13.86	14/101
PhD	3.96	4/101
**Income (£)**		
<10,000	13.4	13/97
10,000–19,000	7.22	7/97
20,000–29,000	13.4	13/97
30,000–39,900	10.31	10/97
40,000–49,900	11.34	11/97
>50,000	44.33	43/97
**Ethnicity**		
White	74.51	76/102
Asian	13.73	14/102
Black/Afro-Caribbean	2.94	3/102
Mixed	6.86	7/102
Other	1.96	2/102
**Religion**		
Christian	59.41	60/101
Muslim	6.93	7/101
Jewish	4.95	5/101
Buddhist	0.99	1/101
Hindu	1.98	2/101
Sikh	1.98	2/101
No religion	22.77	23/101
Other	0.99	1/101
**FH/clinical criteria positive**	0	0/102
**FH of cancer**		
**Total number of participants with any FH of ovarian cancer**	11.76	12/102
Number of participants with a FDR with ovarian cancer	5.88	6/102
Number of participants with a SDR with ovarian cancer	5.88	6/102
Number of participants with a TDR with ovarian cancer	0	0/102
**Total number of participants with any FH of breast cancer**	44.12	45/102
Number of participants with a FDR with breast cancer	14.71	15/102
Number of participants with a SDR with breast cancer	23.53	24/102
Number of participants with a TDR with breast cancer	5.88	6/102
**Total number of participants with a FH of breast and ovarian cancer**	0.98	1/102
**Total number of participants with any FH of prostate cancer**	17.65	18/102
Number of participants with a FDR with prostate cancer	7.84	8/102
Number of participants with a SDR with prostate cancer	7.84	8/102
Number of participants with a TDR with prostate cancer	1.96	2/102
**Total number of participants with any FH of pancreatic cancer**	4.9	5/102
Number of participants with a FDR with pancreatic cancer	0.00	0/102
Number of participants with a SDR with pancreatic cancer	3.92	4/102
Number of participants with a TDR with pancreatic cancer	0.98	1/102
**Psychiatric history**		
Depression	9	9/100
Other psychiatric condition	5	5/100
Current medication for psychiatric condition	5	5/100
**Personal history of non-ovarian cancer**		
Breast cancer	0	0/102
Other cancers	3.92	4/102
**Previous genetic testing unrelated to HBOC**	1.98	2/101
**Breast self-examination in the last 12 months**		
Never	29.41	30/102
<once a month	48.04	49/102
once a month	14.71	15/102
>once a month	7.84	8/102
**Clinical screening for breast cancer**		
Ever had a clinical breast exam	56.57	56/99
Ever had a MRI	4	4/100
Ever had a mammogram	54.46	55/101
**Ovarian cancer screening**		
Currently undergoing screening	1.96	2/102
Have previously undergone screening	11.11	11/99
**Previous surgical prevention to prevent ovarian cancer**	0	0/102
**Health behaviour and lifestyle**		
**Smoking**		
Ever smokers	25.49	26/102
Current smokers	7.07	7/99
**Alcohol consumption in the past 12 months**		
Every week	48.04	49/102
Every month	14.71	15/102
Less frequently than once a month	21.57	22/102
Not at all	15.69	16/102
Median alcohol consumption on a typical day in units (IQR)		2 (1–2)
**≥5 portions of fruit and vegetables**	63.64	63/99
**Number of participants who consume red meat**	81.37	83/102
**Number of participants currently using vitamin supplements**	51.51	51/99
**Physical exercise (past month)**	78.43	80/102
**Risk prediction algorithm results**		
High lifetime ovarian cancer risk	0.97	1/103
Intermediate ovarian cancer risk	0	0/103
Low lifetime ovarian cancer risk	97.09	100/103
Excluded *	1.94	2/103
**Mean lifetime risk prediction score (excluding the high-risk participant (SD, range))**		1.39 (0.69, 0.56–4.38)
**Mean lifetime risk prediction score (including the high-risk participant (SD, range))**		1.80 (4.10, 0.56–41.98)

FH: family history; FDR: first-degree relative; SDR: second-degree relative; TDR: third-degree relative; HBOC: hereditary breast and ovarian cancer; SD: standard deviation. * One participant was excluded because she entered this study at age 84 and the model predicts risks to age 80. A second participant was excluded because she did not provide any baseline demographic information. Both participants were provided with their high-penetrance gene results, but personalised risk scores were not provided.

**Table 2 cancers-12-01241-t002:** Decision aid evaluation questionnaire responses.

Satisfaction			%				N			
Very satisfied			47.06				48/102			
Satisfied			45.1				46/102			
Neither satisfied nor dissatisfied			5.88				6/102			
Dissatisfied			0.98				1/102			
Very dissatisfied			0.98				1/102			
**Amount of information provided**										
Too little			1.96				2/102			
About right			98.04				100/102			
Too much			0				0/102			
**Length of time taken to view DA**										
Too short			0				0/102			
About right			97.06				99/102			
Too long			2.94				3/102			
**Do any parts of the DA require more detail?**										
Yes			2.97				3/101			
No			97.03				98/101			
**Are there any parts of the DA that should be left out?**										
Yes			0				0/99			
No			100				99/99			
**Would you recommend DA use?**										
Yes			82.18				83/101			
No			0.99				1/101			
Not sure			16.83				17/101			
**How much the DA improved understanding of:**	**Not at all**		**Not very much**		**Somewhat**		**Quite a bit**		**A lot**	
	**%**	**N**	**%**	**N**	**%**	**N**	**%**	**N**	**%**	**N**
OC	7.92	8/101	12.87	13/101	34.65	35/101	28.71	29/101	15.84	16/101
Disadvantages of discovering OC risk (%)	3.96	4/101	6.93	7/101	40.59	41/101	28.71	29/101	19.8	20/101
Advantages of discovering OC risk	2.97	3/101	7.92	8/101	26.73	27/101	36.63	37/101	25.74	26/101
Genetic testing for OC genes	1.98	2/101	8.91	9/101	29.7	30/101	33.66	34/101	25.74	26/101
Implications of carrying OC gene alteration	3.96	4/101	5.94	6/101	36.63	37/101	28.71	29/101	24.75	25/101
**Emotional response to DA**										
Worried/concerned	56.44	57/101	27.72	28/101	13.86	14/101	1.98	2/101	0	0/101
Reassured	6.86	7/102	12.75	13/102	35.29	36/102	25.49	26/102	19.61	20/102
Upset	80.2	81/101	14.85	15/101	3.96	4/101	0	0/101	0.99	1/101

**Table 3 cancers-12-01241-t003:** Telephone helpline evaluation questionnaire responses.

Telephone Helpline Evaluation Questionnaire	%	N
**Total number of women using the helpline**	12.62	13/103
**Number of participants using helpline who consented to this study**	100	13/13
**Mean number of times used (SD, range)**		1.38 (1.12, 1–5)
**Reason for helpline use**		
To help decide whether to take part in this study	12.5	1/8
To ask a study specific question not related to decision making	75	6/8
Technical assistance with the decision aid	50	4/8
Pregnancy related query	25	2/8
Results query	12.5	1/8
**Satisfaction with helpline**		
Very satisfied	75	6/8
Satisfied	25	2/8
Neither satisfied nor dissatisfied	0	0/8
Dissatisfied	0	0/8
Very dissatisfied	0	0/8
**Did the helpline help with decision making?**		
Yes	37.5	3/8
No	50	4/8
Not sure	12.5	1/8
**Would you recommend helpline use?**		
Yes	75	6/8
No	0	0/8
Not sure	25	2/8

SD: standard deviation; 8/13 participants who used the telephone helpline completed the telephone helpline questionnaire. Data are presented for these eight participants.

**Table 4 cancers-12-01241-t004:** Mean questionnaire scores at baseline and at seven days, three months and six months follow up.

Validated Questionnaire	Baseline	7 Days Post Results	3 Months Post Results	6 Months Post Results
HADS				
Total	9.06 (SD = 6.11, range 0–23)	10.43 (SD = 6.26, range 0–30)	9.78 (SD = 7.1, range 0–31)	9.64 (SD = 7.06, range 0–28)
Anxiety	6.11 (SD = 4.05, range 0–17)	7.02 (SD = 4.02, range 0–18)	6.35 (SD = 3.97, range 0–16)	6.1 (SD = 4.06, range 0–15)
Depression	2.92 (SD = 2.9, range 0–11)	3.41 (SD = 3.07, range 0–14)	3.36 (SD = 3.71, range 0–16)	3.55 (SD = 3.62, range 0–14)
EQ-5D-5L				
Total	0.86 (SD = 0.14, range 0.382–1)	0.84 (SD = 0.17, range 0.259–1)	0.83 (SD = 0.21, range 0.051–1)	0.84 (SD = 0.17, range –0.035–1)
VAS	81.27 (SD = 13.9, range 35–100)	80.61 (SD = 16.11, range 4–100)	80.45 (SD = 18.81, range 15–100)	80.76 (SD = 15.3, range 20–100)
Mobility	1.25 (SD = 0.57, range 1–3)	1.26 (SD = 0.55, range 1–3)	1.35 (SD = 0.8, range 1–5)	1.47 (SD = 0.92, range 1–5)
Self-care	1.08 (SD = 0.39, range 1–4)	1.04 (SD = 0.2, range 1–2)	1.12 (SD = 0.57, range 1–5)	1.21 (SD = 0.82, range 1–5)
Usual activities	1.25 (SD = 0.57, range 1–4)	1.24 (SD = 0.52, range 1–3)	1.38 (SD = 0.77, range 1–5)	1.42 (SD = 0.94, range 1–5)
Pain/discomfort	1.55 (SD = 0.71, range 1–4)	1.65 (SD = 0.8, range 1–4)	1.68 (SD = 0.95, range 1–5)	1.71 (SD = 0.82, range 1–5)
Anxiety/depression	1.43 (SD = 0.69, range 1–4)	1.58 (SD = 0.88, range 1–5)	1.58 (SD = 0.88, range 1–5)	1.53 (SD = 0.71, range 1–3)
IES		7.93 (SD = 15.06, range 0–67)	7.57 (SD = 17.07, range 0–73)	4.95 (SD = 10.61, range 0–48)
CWS	5.8 (SD = 1.96, range 4–14)	5.13 (SD = 1.61, range 4–12)	5.04 (SD = 1.51, range 4–11)	5.17 (SD = 1.61, range 4–11)
CRP				
CRP Likert scale	2.93 (SD = 0.78, range 1–5)	2.72 (SD = 0.83, range 1–5)	2.86 (SD = 0.66, range 1–5)	2.93 (SD = 0.74, range 1–5)
CRP VAS	46.05 (SD = 22.1, range 0–90)	44.51 (SD = 24.61, range 2–90)	47.43 (SD = 21.81, range 0–90)	49.67 (SD = 22.84, 1–90)
DRS				
DRS scale			27 (SD = 52.11, range 0–250)	
DRS Madalinska			1.16 (SD = 0.4, range 1–3)	

HADS: Hospital Anxiety and Depression Scale questionnaire; IES: Impact of Events Scale questionnaire; CRP: Cancer Risk Perception questionnaire; CWS: Cancer Worry Scale questionnaire; DRS: Decision Regret Satisfaction questionnaire; VAS: Visual Analogue Scale. HADS: 14 item questionnaire, with 7 items pertaining to anxiety and 7 to depression. Each item is scored on a 4-point Likert scale, from 0 to 3, with total scores ranging from 0 to 42. Higher scores indicate higher levels of anxiety/depression. EQ-5D-5L: EuroQol-5D-5L 5 item questionnaire. Each item (mobility, self-care, usual activities, pain/discomfort, and anxiety/depression) is scored on a 5-point Likert scale, from 1 to 5. Higher scores indicate poorer health. Total scores are then converted into a utility value using published reference values for the UK by the EuroQol Research Foundation. Utility values range from 0 to 1, with 0 indicating the worst health and 1 the best health. In addition, participants are asked to state “how good or bad your health is today” using a Visual Analogue Scale ranging from 0 (worst health) to 100 (best health). IES: 15 item questionnaire. Each item is scored on a 4-point Likert scale, from 0 to 5, with total scores ranging from 0 to 75. Higher scores indicate higher distress levels. CWS: 4 item questionnaire. Each item is scored on a 4-point Likert scale, from 1 to 4, with total scores ranging from 4 to 16. Higher scores indicate greater worry of developing ovarian cancer. CRP: 1 item questionnaire. The item is scored on a 5-point Likert scale, from 1 to 5. A higher score indicates that the individual perceives that they are at greater risk of developing cancer of any type at some point in their life compared to other women of the same age. In addition, participants are asked to state “on a scale from 0 to 100, where 0 is no chance at all, and 100 is absolutely certain, what do you think are the chances that you will get cancer (of any type) sometime during your lifetime?” DRS: First part consists of a 5 item questionnaire (Decision Satisfaction Regret Scale. Each item is scored on a 5-point Likert scale, from 0 to 100, with total scores ranging from 0 to 500. Higher scores indicate less satisfaction/more regret. Second part consists of a 1 item questionnaire (Madalinska). The item is scored on a 5-point Likert scale, from 0 to 100. Higher scores indicate less satisfaction/more regret.

**Table 5 cancers-12-01241-t005:** Linear random-effects mixed models and overall contrast tests for study outcomes.

Model and Variable	Coef.	Std. Err	p > |z|	95% CI
**HADS Total**				
FH Breast Cancer	1.66	1.334	0.217	−0.782 to 4.3
FH Ovarian Cancer	0.426	2.146	0.843	−3.766 to 4.616
* 7 Days	1.068	0.513	0.039	0.159 to 2.071
3 Months	0.986	0.537	0.068	−0.094 to 2.148
* 6 Months	1.268	0.557	0.024	0.215 to 2.385
Age	−0.015	0.046	0.742	−0.101 to 0.078
Income	−0.633	0.411	0.127	−1.452 to 0.178
Marital Status	1.321	1.424	0.356	−1.608 to 4.675
Ethnicity	3.349	1.899	0.081	−0.716 to 7.141
Religion: Jewish	−1.715	3.116	0.584	−7.559 to 4.456
Religion: Muslim	−6.223	3.228	0.057	−12.655 to 0.751
Religion: Atheist	−0.736	1.582	0.643	−3.893 to 2.478
Religion: Other	−3.759	2.848	0.19	−9.587 to 2.509
**HADS Total**	**df**	**Chi-sq**	***p*-value**	
**^#^ BL vs. Overall (joint)**	3	5.2	0.158	
**HADS Anxiety**				
FH Breast Cancer	0.933	0.801	0.248	−0.63 to 2.625
FH Ovarian Cancer	−0.142	1.285	0.912	−2.552 to 2.373
* 7 Days	0.649	0.326	0.048	0.006 to 1.278
3 Months	0.339	0.339	0.318	−0.377 to 1.022
6 Months	0.172	0.353	0.628	−0.498 to 0.88
Age	−0.026	0.028	0.346	−0.083 to 0.028
Income	−0.216	0.247	0.383	−0.66 to 0.255
Marital Status	0.505	0.856	0.557	−1.285 to 2.123
Ethnicity	1.688	1.143	0.143	−0.433 to 3.799
Religion: Jewish	0.039	1.871	0.984	−3.838 to 3.558
* Religion: Muslim	−4.136	1.941	0.036	−7.79 to −0.387
Religion: Atheist	−0.89	0.951	0.352	−2.71 to 1.051
Religion: Other	−1.975	1.707	0.251	−5.541 to 1.271
**HADS Anxiety**	**df**	**Chi-sq**	***p*-value**	
**^#^ BL vs. Overall (joint)**	3	6.22	0.102	
**HADS Depression**				
FH Breast Cancer	0.778	0.655	0.238	−0.55 to 2.141
FH Ovarian Cancer	0.942	1.086	0.387	−1.201 to 3.16
7 Days	0.481	0.292	0.101	−0.03 to 1.048
* 3 Months	0.68	0.304	0.027	0.062 to 1.257
* 6 Months	1.155	0.317	<0.001	0.54 to 1.772
Age	0.012	0.022	0.586	−0.033 to 0.054
* Income	−0.403	0.2	0.047	−0.805 to −0.031
Marital Status	0.8	0.702	0.258	−0.579 to 2.132
Ethnicity	1.642	0.925	0.079	−0.249 to 3.377
Religion: Jewish	−1.782	1.53	0.248	−5.021 to 1.214
Religion: Muslim	−1.996	1.562	0.205	−5.08 to 1.209
Religion: Atheist	0.148	0.776	0.85	−1.312 to 1.787
Religion: Other	−1.769	1.388	0.206	−4.607 to 1.095
**HADS Depression**	**df**	**Chi-sq**	***p*-value**	
**^#^ BL vs. Overall (joint)**	3	3.7	0.296	
**EQ-5D-5L Total**				
FH Breast Cancer	−0.068	0.031	0.034	−0.135 to −0.008
FH Ovarian Cancer	−0.066	0.052	0.204	−0.171 to 0.028
7 Days	−0.026	0.015	0.082	−0.056 to 0.003
* 3 Months	−0.034	0.015	0.025	−0.062 to −0.004
6 Months	−0.033	0.016	0.036	−0.069 to −0.005
Age	−0.002	0.001	0.138	−0.004 to 0.001
Income	0.013	0.01	0.19	−0.007 to 0.032
Marital Status	−0.01	0.034	0.756	−0.076 to 0.061
Ethnicity	−0.05	0.044	0.262	−0.136 to 0.038
Religion: Jewish	−0.011	0.073	0.885	−0.155 to 0.136
Religion: Muslim	0.059	0.075	0.435	−0.106 to 0.209
Religion: Atheist	0.058	0.037	0.12	−0.012 to 0.136
Religion: Other	0.078	0.066	0.242	−0.058 to 0.202
**EQ-5D-5L Total**	**df**	**Chi-sq**	***p*-value**	
**^#^ BL vs. Overall (joint)**	3	0.14	0.987	
**EQ-5D-5L VAS**				
FH Breast Cancer	−4.74	2.792	0.093	−10.024 to 0.915
FH Ovarian Cancer	−6.613	4.908	0.18	−16.246 to 3.636
7 Days	−1.537	1.68	0.361	−5.063 to 2.159
3 Months	−3.244	1.749	0.065	−6.533 to 0.491
6 Months	−2.492	1.798	0.167	−6.161 to 1.046
Age	0.048	0.097	0.621	−0.143 to 0.243
* Income	1.819	0.859	0.037	0.267 to 3.524
Marital Status	−0.386	2.997	0.898	−6.607 to 5.465
Ethnicity	−7.129	4.004	0.078	−15.297 to −0.015
Religion: Jewish	−1.73	6.474	0.79	−14.671 to 10.225
Religion: Muslim	10.205	6.774	0.136	−3.622 to 25.287
Religion: Atheist	0.921	3.322	0.782	−6.351 to 7.29
Religion: Other	2.139	5.9	0.718	−9.416 to 14.159
**EQ-5D-5L VAS**	**df**	**Chi-sq**	***p*-value**	
**^#^ BL vs. Overall (joint)**	3	1.63	0.654	
**IES**				
FH Breast	−2.691	2.783	0.337	−8.768 to 2.934
FH Ovarian	−2.838	4.625	0.541	−11.282 to 6.296
3 Months	0.541	1.688	0.749	−2.841 to 3.86
* 6 Months	−3.533	1.724	0.042	−6.764 to −0.456
Age	−0.009	0.1	0.93	−0.183 to 0.178
Income	−1.53	0.895	0.091	−3.193 to 0.129
Marital Status	−1.252	3.005	0.678	−7.493 to 4.494
Ethnicity	1.551	4.379	0.724	−6.956 to 10.635
Religion: Jewish	7.084	6.242	0.26	−5.696 to 18.636
Religion: Muslim	−12.871	7.211	0.078	−25.685 to 2.043
* Religion: Atheist	−8.159	3.377	0.018	−15.396 to −1.292
Religion: Other	−3.833	6.028	0.527	−16.369 to 8.471
**IES**	**df**	**Chi-sq**	***p*-value**	
**^#^ BL vs. Overall (joint)**	2	2.78	0.249	
**CWS**				
FH Breast Cancer	0.019	0.306	0.952	−0.616 to 0.639
FH Ovarian Cancer	−0.138	0.535	0.797	−1.316 to 1.001
* 7 Days	−0.73	0.182	<0.001	−1.096 to −0.399
* 3 Months	−0.802	0.189	<0.001	−1.173 to −0.448
* 6 Months	−0.775	0.195	<0.001	−1.16 to −0.358
Age	−0.013	0.011	0.215	−0.035 to 0.009
Income	−0.126	0.094	0.186	−0.325 to 0.065
Marital Status	0.335	0.329	0.311	−0.374 to 0.973
Ethnicity	0.585	0.44	0.187	−0.264 to 1.439
Religion: Jewish	1.375	0.71	0.056	−0.091 to 2.772
Religion: Muslim	−1.073	0.737	0.149	−2.593 to 0.455
Religion: Atheist	−0.491	0.365	0.182	−1.224 to 0.247
Religion: Other	1.157	0.647	0.078	−0.035 to 2.428
**CWS**	**df**	**Chi-sq**	***p*-value**	
**^#^ BL vs. Overall (joint)**	3	9.7	0.021	
**CRP Likert Scale**				
FH Breast Cancer	0.436	0.138	0.002	0.135 to 0.696
FH Ovarian Cancer	0.248	0.238	0.299	−0.207 to 0.748
* 7 Days	−0.195	0.077	0.012	−0.349 to −0.041
3 Months	−0.093	0.079	0.241	−0.251 to 0.068
6 Months	−0.062	0.083	0.454	−0.237 to 0.111
Age	−0.007	0.005	0.131	−0.017 to 0.002
Income	0.024	0.042	0.58	−0.071 to 0.107
Marital Status	−0.032	0.148	0.83	−0.308 to 0.269
Ethnicity	−0.089	0.197	0.654	−0.475 to 0.309
Religion: Jewish	0.173	0.321	0.592	−0.475 to 0.73
Religion: Muslim	−0.179	0.332	0.59	−0.89 to 0.524
Religion: Atheist	−0.115	0.164	0.486	−0.44 to 0.196
Religion: Other	0.127	0.292	0.665	−0.459 to 0.714
**CRP Likert Scale**	**df**	**Chi-sq**	***p*-value**	
**^#^ BL vs. Overall (joint)**	3	10.44	0.015	
**CRP VAS**				
FH Breast Cancer	6.455	4.416	0.148	−2.004 to 15.098
FH Ovarian Cancer	−4.31	7.495	0.566	−19.607 to 9.766
7 Days	−2.985	2.441	0.223	−7.692 to 2.013
3 Months	0.579	2.675	0.829	−4.12 to 5.918
6 Months	−0.093	2.77	0.973	−5.488 to 5.632
Age	−0.194	0.153	0.207	−0.487 to 0.084
Income	1.471	1.353	0.28	−1.386 to 4.059
Marital Status	0.794	4.727	0.867	−9.788 to 9.475
Ethnicity	−9.792	6.261	0.121	−22.644 to 3.523
Religion: Jewish	3.429	10.238	0.739	−15.525 to 24.542
Religion: Muslim	−0.805	10.561	0.939	−22.586 to 20.999
Religion: Atheist	−6.437	5.263	0.225	−16.92 to 3.543
Religion: Other	1.36	9.434	0.886	−17.183 to 21.316
**CRP VAS**	**df**	**Chi-sq**	***p*-value**	
**^#^ BL vs. Overall (joint)**	3	2.51	0.474	
**DRS scale**				
FH Breast Cancer	1.741	12.287	0.888	−22.846 to 26.328
FH Ovarian Cancer	11.808	17.974	0.514	−24.157 to 47.773
Age	0.208	0.437	0.636	−0.667 to 1.083
Income	−0.001	<0.001	0.216	−0.001 to 0
Marital Status	24.362	14.121	0.09	−3.894 to 52.617
* Ethnicity	47.091	20.396	0.024	6.28 to 87.902
Religion: Jewish	24.538	25.272	0.336	−26.031 to 75.108
Religion: Muslim	−11.145	32.366	0.732	−75.909 to 53.62
Religion: Atheist	10.411	16.022	0.518	−21.65 to 42.471
* Religion: Other	62.958	25.021	0.015	12.891 to 113.024
**DRS Madalinska**				
FH Breast Cancer	0.118	0.093	0.21	−0.068 to 0.303
FH Ovarian Cancer	0.022	0.136	0.871	−0.25 to 0.294
Age	0.001	0.003	0.849	−0.006 to 0.007
* Income	<0.001	<0.001	0.045	0 to 0
* Marital Status	0.247	0.107	0.025	0.033 to 0.46
Ethnicity	0.18	0.154	0.248	−0.129 to 0.488
Religion: Jewish	0.145	0.191	0.451	−0.238 to 0.528
Religion: Muslim	−0.151	0.245	0.541	−0.641 to 0.34
Religion: Atheist	0.068	0.121	0.573	−0.173 to 0.31
* Religion: Other	0.642	0.189	0.001	0.263 to 1.021

HADS: Hospital Anxiety and Depression Scale questionnaire; IES: Impact of Events Scale questionnaire; CRP: Cancer Risk Perception questionnaire; CWS: Cancer Worry Scale questionnaire; DRS: Decision Regret Satisfaction questionnaire; VAS: Visual Analogue Scale; FH: family history; Coef: coefficient; Std. Err: standard error. FH of breast cancer: positive versus negative (reference category); FH of ovarian cancer: positive versus negative (reference category); 7 days: questionnaire scores at 7 days versus baseline (reference category); 3 months: questionnaire scores at 3 months versus baseline (reference category); 6 months: questionnaire scores at 6 months versus baseline (reference category); age: age in years (continuous variable); income: as a continuous variable, but measured in £10,000 increments; marital status: cohabiting/living with partner/married (reference category) versus divorced/separated/single/widowed; ethnicity: Caucasian (reference category) versus non-Caucasian; religion Jewish: Christian (reference category) versus Jewish; religion Muslim: Christian (reference category) versus Muslim; Atheist: Christian (reference category) versus atheist; religion other (Hindu, Buddhist, Sikh): Christian (reference category) versus other. ^#^ BL vs. Overall (joint): Overall contrast test reflecting whether the mean outcome scale values at each time point (7 days, 3 months, or 6 months) were jointly different from the baseline for the whole group. This showed a significant decrease for CWS and CRP (Likert), but no significant change for HADS, HADS anxiety, HADS depression, EQ-5D-5L, IES and CRP (VAS) outcomes jointly over time. * Variables of statistical significance (*p* < 0.05).

**Table 6 cancers-12-01241-t006:** Decision Regret Satisfaction questionnaire responses according to individual questionnaire items.

Questionnaire Items	Questionnaire Responses									
	Strongly Disagree		Disagree		Neither Agree nor Disagree		Agree		Strongly Agree	
	%	N	%	N	%	N	%	N	%	N
**It was the right decision**	0	0/76	0	0/76	1.32	1/76	13.16	10/76	85.53	65/76
**I regret the choice that was made**	80.26	61/76	14.47	11/76	2.63	2/76	1.32	1/76	1.32	1/76
**I would go for the same choice if I had to do it over again**	0	0/76	1.32	1/76	3.95	3/76	13.16	10/76	81.58	62/76
**The choice did me a lot of harm**	89.33	67/75	8	6/75	2.67	2/75	0	0/75	0	0/75
**The decision was a wise one**	1.32	1/76	0	0/76	2.63	2/76	13.16	10/76	82.89	63/76
**I am satisfied with the decision I have made**	0	0/76	0	0/76	1.32	1/76	13.16	10/76	85.53	65/76

**Table 7 cancers-12-01241-t007:** Contrast tests for between-group and within-group analyses over time.

**HADS total**	**df**	**Chi-sq**	***p*-value**	**CRP Likert scale**	**df**	**Chi-sq**	***p*-value**
**Event#Group**				**Event#Group**			
BL vs. 7 Days (joint)	1	0.43	0.51	BL vs. 7 Days (joint)	1	1.16	0.281
BL vs. 3 Months (joint)	1	0.07	0.797	BL vs. 3 Months (joint)	1	2.7	0.101
BL vs. 6 Months (joint)	1	0.13	0.719	BL vs. 6 Months (joint)	1	0.78	0.378
BL vs. Overall (joint)	3	1.05	0.788	BL vs. Overall (joint)	3	2.8	0.423
**Event|Group**				**Event|Group**			
BL vs. Joint|OC FH−	3	7.08	0.07	BL vs. Joint|OC FH−	3	6.69	0.083
BL vs. Joint|OC FH+	3	2.57	0.463	BL vs. Joint|OC FH+	3	6.59	0.086
**HADS anxiety**	**df**	**Chi-sq**	***p*-value**	**CRP VAS**	**df**	**Chi-sq**	***p*-value**
**Event#Group**				**Event#Group**			
BL vs. 7 Days (joint)	1	0.62	0.431	BL vs. 7 Days (joint)	1	2.23	0.135
BL vs. 3 Months (joint)	1	<0.01	0.972	BL vs. 3 Months (joint)	1	0.51	0.477
BL vs. 6 Months (joint)	1	0.04	0.849	BL vs. 6 Months (joint)	1	1.1	0.294
BL vs. Overall (joint)	3	1.05	0.79	BL vs. Overall (joint)	3	5.63	0.131
**Event|Group**				**Event|Group**			
BL vs. Joint|OC FH−	3	4.39	0.222	BL vs. Joint|OC FH−	3	2.37	0.5
BL vs. Joint|OC FH+	3	3.51	0.319	BL vs. Joint|OC FH+	3	4.37	0.224
**HADS depression**	**df**	**Chi-sq**	***p*-value**	**EQ-5D-5L UK score**	**df**	**Chi-sq**	***p*-value**
**Event#Group**				**Event#Group**			
BL vs. 7 Days (joint)	1	0.03	0.869	BL vs. 7 Days (joint)	1	2.36	0.125
BL vs. 3 Months (joint)	1	<0.01	0.971	BL vs. 3 Months (joint)	1	2.44	0.118
BL vs. 6 Months (joint)	1	0.91	0.339	BL vs. 6 Months (joint)	1	2.25	0.133
BL vs. Overall (joint)	3	1.18	0.759	BL vs. Overall (joint)	3	3.66	0.3
**Event|Group**				**Event|Group**			
* BL vs. Joint|OC FH−	3	14.18	0.003	BL vs. Joint|OC FH−	3	7	0.072
BL vs. Joint|OC FH+	3	0.73	0.866	BL vs. Joint|OC FH+	3	1.11	0.774
**IES Score**	**df**	**Chi-sq**	***p*-value**	**EQ-5D-5L VAS**	**df**	**Chi-sq**	***p*-value**
**Event#Group**				**Event#Group**			
BL vs. 3 Months (joint)	1	1.09	0.297	BL vs. 7 Days (joint)	1	0.64	0.425
BL vs. 6 Months (joint)	1	0.01	0.93	* BL vs. 3 Months (joint)	1	6.47	0.011
BL vs. Overall (joint)	2	1.3	0.523	BL vs. 6 Months (joint)	1	1.4	0.237
**Event|Group**				BL vs. Overall (joint)	3	6.74	0.081
* BL vs. Joint|OC FH−	2	6.31	0.043	**Event|Group**			
BL vs. Joint|OC FH+	2	1.29	0.524	BL vs. Joint|OC FH−	3	3.89	0.273
**CWS Score**	**df**	**Chi-sq**	***p*-value**				
**Event#Group**							
BL vs. 7 Days (joint)	1	0.99	0.32				
BL vs. 3 Months (joint)	1	0.25	0.615				
BL vs. 6 Months (joint)	1	0.18	0.675				
BL vs. Overall (joint)	3	1	0.802				
**Event|Group**							
* BL vs. Joint|OC FH−	3	26.92	<0.001				
BL vs. Joint|OC FH+	3	1.95	0.582				

BL: baseline; FH: family history; HADS: Hospital Anxiety and Depression Scale questionnaire; IES: Impact of Events Scale questionnaire; CRP: Cancer Risk Perception questionnaire; CWS: Cancer Worry Scale questionnaire; VAS: Visual Analogue Scale. ‘Group’ refers to either participants with a family history of ovarian cancer (OC FH+ group) or no family history of ovarian cancer (OC FH− group). ‘Event#Group’ refers to the group–time interaction, which reflects the ‘between-group’ (OC FH+ vs. OC FH−) difference over time. BL vs. 7 days (joint), BL vs. 3 months (joint), and BL vs. 6 months (joint) reflect whether the mean between-group difference at each time point (7 days, 3 months, or 6 months) was different from baseline. BL vs. Overall (joint) reflects whether the mean between-group differences at each time point (7 days, 3 months, or 6 months) were jointly different from the baseline between-group difference. Event|Group refers to the group–time interaction, which reflects the ‘within-group’ difference over time. BL vs. Joint |OC FH− reflects whether the mean outcome scale value at each time point (7 days, 3 months, or 6 months) was jointly different from the baseline within the ovarian cancer family history negative group. BL vs. Joint |OC FH+ reflects whether the mean outcome scale value at each time point (7 days, 3 months, or 6 months) was jointly different from the baseline within the ovarian cancer family history positive group. * Statistical significance (*p* < 0.05).

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
