# Peer review of "Population Study of Ovarian Cancer Risk Prediction for Targeted Screening and Prevention"

_cancers, 2020, doi:10.3390/cancers12051241_

Round 1

Reviewer 1 Report

The authors describe the results of an unselected population-based, prospective cohort study aimed at the evaluation of the lifetime ovarian cancer (OC) risk using a personalized risk-tool calculator followed by risk-management. About 50% of recruited women (n=105, final n=103 following 2 withdrawals) gave their consent to undergo the study after a DA questionnaire. The study included the use of an OC risk-prediction-algorithm that combined the use of lifestyle/hormonal OC risk-factors and genetic testing (BRCA1/BRCA2/RAD51C/RAD51D/BRIP1, OC-susceptibility single-nucleotide-polymorphisms). They assessed also uptake/acceptability of the population-genetic-testing (PGT)/OC risk-stratification, women satisfaction, decision-aid/telephone-helpline use, psychological-health and quality-of-life using validated/customised questionnaires over six-months. They used also linear mixed-models/contrast-tests analysed impact on study outcomes. They conclude that this method represents one of the first evidence about the feasibility of OC-risk-stratification of general population women using an algorithm tool that considers also the risk management.

This study is of interest considering that ovarian cancer is mostly diagnosed at high stages and is lacking of effective screening tests. However, some parts of the study need to be better explained.

Major comments:

Results of NGS assays on BRCA1/BRCA2/RAD51C/RAD51D/BRPI1 genes and the 30 SNPs validated for OC are not fully described either commented in the text, but it seems they are only presented in Table1. In particular, some points should be described clearer. What are the most genetic variations found, if any? Does they reflect the frequency found in the clinic? Are you able to predict how much the presence of one or more genetic variants contribute in the RPA respect to the other factors considered (i.e. age/ OC-FH/ BMI/ HRT/ OCP use/ endometriosis/ parity, as described in lines 390-392)  for the calculation of the OC risk?  These data should be clearer presented and discussed.

In general, it could be useful for a better and clearer understanding of tables to highlight data that are statistically significant.

Minor revision:

Some typing errors are present in the text. Please correct them.

Author Response

Response to Reviewers’ Comments

REVIEWER-1

The authors describe the results of an unselected population-based, prospective cohort study aimed at the evaluation of the lifetime ovarian cancer (OC) risk using a personalized risk-tool calculator followed by risk management. About 50% of recruited women (n=105, final n=103 following 2 withdrawals) gave their consent to undergo the study after a DA questionnaire. The study included the use of an OC risk-prediction-algorithm that combined the use of lifestyle/hormonal OC risk-factors and genetic testing (BRCA1/BRCA2/RAD51C/RAD51D/BRIP1, OC-susceptibility single-nucleotide-polymorphisms). They assessed also uptake/acceptability of the population genetic- testing (PGT)/OC risk-stratification, women satisfaction, decision-aid/telephone-helpline use, psychological-health and quality-of-life using validated/customised questionnaires over six-months. They used also linear mixed-models/contrast-tests analysed impact on study outcomes. They conclude that this method represents one of the first evidence about the feasibility of OC-risk-stratification of general population women using an algorithm tool that considers also the risk management. This study is of interest considering that ovarian cancer is mostly diagnosed at high stages and is lacking of effective screening tests. However, some parts of the study need to be better explained.

Response

We thank the reviewer for his/her comment

Major comments:

Results of NGS assays on BRCA1/BRCA2/RAD51C/RAD51D/BRPI1 genes and the 30 SNPs validated for OC are not fully described either commented in the text, but it seems they are only presented in Table1. In particular, some points should be described clearer. What are the most genetic variations found, if any? Does they reflect the frequency found in the clinic? Are you able to predict how much the presence of one or more genetic variants contribute in the RPA respect to the other factors considered (i.e. age/ OC-FH/ BMI/ HRT/ OCP use/ endometriosis/ parity, as described in lines 390-392) for the calculation of the OC risk? These data should be clearer presented and discussed.

Response

We thank the reviewer for his / her comment. We had modelled risk using the entire RPA. We have now performed the model predictions using two additional versions of the model, after excluding the identified BRCA1 mutation carrier: (1) including the PRS only; (2) including only the epidemiological risk factors. In both cases we considered the family history information and the fact that all women did not carry a rare pathogenic variant. We have plotted these remaining lifetime risks in a new figure (Supplementary Figure-1) along with the full model predictions. As the figure (Supplementary Figure-1) demonstrates, the epidemiological risk factors provide a greater level of ovarian cancer risk stratification among the participants compared to the PRS alone. The variability in the predicted risks increases further when the full model is considered. These patterns are in line with the expected (theoretical) model behaviour where for ovarian cancer, greater levels of risk stratification are expected in the population on the basis of epidemiological risk factors alone compared to a model considering only the PRS. However, further improvement in risk stratification can be obtained when both the PRS and risk factors are considered jointly (manuscript in preparation). We have now included the figure in the Supplementary material and included the following sentences in the text (lines 111-115):

“As expected under the algorithm, the epidemiological risk factors  alone provide a greater level of ovarian cancer risk stratification among the participants compared to the PRS alone (Supplementary Figure S1). However, risk stratification is further improved when the full model incorporating both epidemiological risk factors and PRS is considered.”

In general, it could be useful for a better and clearer understanding of tables to highlight data that are statistically significant.

Response

We thank the reviewer for his/her comment. We have edited the tables to better highlight the data which are significant. The explanation at the footnote has also been expanded. It is important to point out that lack of significance for some outcomes is also an important finding and being focused only on statistical significance (p<0.05) may not be the best inference for some outcomes. Additionally the effect size is also important.

Minor revision:

Some typing errors are present in the text. Please correct them.

Response

Typographical errors have been corrected in text. Thank you for pointing this out.

Reviewer 2 Report

This is a study of a OC-risk-stratification of general-population women using a
personalised-OC risk-tool followed by risk-management.

Access to genetic testing is restricted, so such questionnaire based stratification may be useful to proper target individuals for testing. Limitation of the study is a small sample and relatively short follow-up. However, this study showes the direction for further large studies with long follow-up.

One remark: there is a lot of data in large tables, quite difficult to read and understand.

Author Response

REVIEWER-2

This is a study of a OC-risk-stratification of general population women using a personalised-OC risk-tool followed by risk-management. Access to genetic testing is restricted, so such questionnaire based stratification may be useful to proper target individuals for testing. Limitation of the study is a small sample and relatively short follow-up. However, this study shows the direction for further large studies with long follow-up.

One remark: there is a lot of data in large tables, quite difficult to read and understand

Response

We thank the reviewer for his/her comment

As described in response to reviewer-1's comments, we have edited the tables to better highlight the data which are significant. We hope this will make it easier to read.